Quantification of human enteric viruses as alternative indicators of fecal pollution to evaluate wastewater treatment processes

Garcia Audrey 1
Le Tri 1
Jankowski Paul 1
Yanaç Kadir 2
Yuan Qiuyan 2
Uyaguari-Diaz Miguel I. Miguel.Uyaguari@umanitoba.ca 1
1 Department of Microbiology, University of Manitoba , Winnipeg , Manitoba , Canada
2 Department of Civil Engineering, University of Manitoba , Winnipeg , Manitoba , Canada
Grande-Pérez Ana
Electronic publication date: 2022 Feb 14
Publication date: 2022
Volume: 10
Electronic Location ID: e12957
Received 2021 Aug 6; Accepted 2022 Jan 27
Copyright: ©2022 Garcia et al.
Copyright year: 2022
Copyright holder: Garcia et al.
License: This is an open access article distributed under the terms of the Creative Commons Attribution License, which permits unrestricted use, distribution, reproduction and adaptation in any medium and for any purpose provided that it is properly attributed. For attribution, the original author(s), title, publication source (PeerJ) and either DOI or URL of the article must be cited.
License URL: https://creativecommons.org/licenses/by/4.0/

Keywords: Enteric viruses, Wastewater, Adenovirus, crAssphage, Pepper mild mottle virus, uidA, Viral markers, Escherichia coli, Quantitative PCR

Funding: Miguel Uyaguari-Diaz at the University of Manitoba 322388 The Faculty of Science, University of Manitoba 322788 Research start-up funds grant No. 322388 were assigned to Miguel Uyaguari-Diaz at the University of Manitoba. Collaborative grant No. 322788 (Drs. Uyaguari and Yuan) was awarded by the Faculty of Science, University of Manitoba. The funders had no role in study design, data collection and analysis, decision to publish, or preparation of the manuscript.

==============================
We investigated the potential use and quantification of human enteric viruses in municipal wastewater samples of Winnipeg (Manitoba, Canada) as alternative indicators of contamination and evaluated the processing stages of the wastewater treatment plant. During the fall 2019 and winter 2020 seasons, samples of raw sewage, activated sludge, effluents, and biosolids (sludge cake) were collected from the North End Sewage Treatment Plant (NESTP), which is the largest wastewater treatment plant in the City of Winnipeg. DNA (Adenovirus and crAssphage) and RNA enteric viruses (Pepper mild mottle virus, Norovirus genogroups GI and GII, Rotavirus Astrovirus, and Sapovirus) as well as the uidA gene found in Escherichia coli were targeted in the samples collected from the NESTP. Total nucleic acids from each wastewater treatment sample were extracted using a commercial spin-column kit. Enteric viruses were quantified in the extracted samples via quantitative PCR using TaqMan assays. Overall, the average gene copies assessed in the raw sewage were not significantly different (p-values ranged between 0.1023 and 0.9921) than the average gene copies assessed in the effluents for DNA and RNA viruses and uidA in terms of both volume and biomass. A significant reduction (p-value ≤ 0.0438) of Adenovirus and Noroviruses genogroups GI and GII was observed in activated sludge samples compared with those for raw sewage per volume. Higher GCNs of enteric viruses were observed in dewatered sludge samples compared to liquid samples in terms of volume (g of sample) and biomass (ng of nucleic acids). Enteric viruses found in gene copy numbers were at least one order of magnitude higher than the E. coli marker uidA, indicating that enteric viruses may survive the wastewater treatment process and viral-like particles are being released into the aquatic environment. Viruses such as Noroviruses genogroups GI and GII, and Rotavirus were detected during colder months. Our results suggest that Adenovirus, crAssphage, and Pepper mild mottle virus can be used confidently as complementary viral indicators of human fecal pollution.

Introduction

The human fecal waste present in raw sewage (RS) contains pathogens that can cause numerous diseases. This can have a huge negative impact to public, aquatic health, and the economy (Stachler et al., 2017). Wastewater treatment plants (WWTPs) serve as protective barriers between communities and the environment by reducing the organic matter present in wastewater. Water quality is currently assessed using traditional markers such as coliforms and Escherichia coli, leaving other microbes such as viruses largely unexplored. The North End Sewage Treatment Plant (NESTP) in Winnipeg, Manitoba handles approximately 70% of the city’s wastewater treatment, serving over 400,000 people (City of Winnipeg, Water and Waste Department, 2020). The treatment process at the NESTP first involves RS undergoing primary treatment to remove solids. During the next treatment cycle, activated sludge (AS), a heterotrophic cocktail of bacteria and protozoa, degrades organic matter present in solid waste. The activated sludge (also known as biological treatment or secondary treatment) is the most widely used process around the world to treat municipal wastewater (Racz, Datta & Goel, 2010; Scholz, 2016), and its use will likely continue due to its low cost and high efficiency. After the biological treatment, wastewater is UV-disinfected and discharged as effluents (EF) into the river (City of Winnipeg, Water and Waste Department, 2020). Approximately 200 million liters of EF are discharged per day (City of Winnipeg, Water and Waste Department, 2020).

The main indicator of biological contamination used in wastewater treatment screening is E. coli, a fecal coliform bacterium (Hood, Ness & Blake, 1983). It is present in the gut of humans and warm-blooded animals, and widely used as the main indicator of fecal pollution during the wastewater treatment process. Nevertheless, the use of only fecal bacteria indicator in wastewater excludes other possible pathogen groups present, such as human enteric viruses. Targeting these viruses in EF could be an alternative method to monitor the wastewater treatment process. Within this context, Dutilh et al. (2014) targeted the DNA crAssphage genome in a human fecal sample. With further bioinformatics testing, it was predicted that the crAssphage genome is highly abundant, having been identified in 73% of human fecal metagenomes surveyed (Dutilh et al., 2014). The dynamics of crAssphage as human source marker has recently been explored in fecally polluted environment (Ballesté et al., 2019; Wu et al., 2020). In terms of RNA viruses, Pepper mild mottle virus (PMMV), a single-stranded RNA plant virus, has been identified highly abundant in feces as a surrogate indicator to assess microbial quality for food, water and wastewater (Zhang et al., 2006; Symonds, Nguyen & Breitbart, 2018).

In the present study, samples of RS, AS, EF, and biosolids/sludge cake (SC) from the NESTP were collected (during fall 2019 and winter 2020) to investigate the potential of quantitating human enteric viruses in wastewater samples as complementary indicators of contamination to evaluate the processing stages of wastewater treatment. DNA enteric viruses in this study include human Adenovirus (AdV) and cross-assembly phage (crAssphage), while RNA enteric viruses include PMMV, Noroviruses (NoV) of the genogroups GI and GII, Astrovirus (AstV), Sapovirus (SaV), and Rotavirus (RoV). We also studied the presence of a molecular marker for E. coli, the uidA gene, in the samples collected from the NESTP. An overview of the workflow is illustrated in Fig. 1.

Materials and Methods

Sample collection

A liter of RS, AS, EF, and 1 kg of SC were collected from the NESTP during each sampling event. Each sample was sealed in a 1-L sterile polyethylene container lined with a sterile plastic bag. Samples were collected on October 22nd, 2019 (Event 1) and November 28th, 2019 (Event 2) in the fall season. In the winter season, samples were collected on December 18th, 2019 (Event 3) and February 6th, 2020 (Event 4). SC samples were collected earlier in the day during Events 3 and 4. All samples were kept at 4 °C and processed within 24 h of collection.

Figure 1 Graphical abstract of workflow.

Ultrafiltration of wastewater samples

Each wastewater treatment sample (RS, AS, and EF), including Millipore Milli-Q water as a negative control, was first filtered via a funnel and cheesecloth to remove any solid waste or debris. Next, 140 mL of each wastewater sample was concentrated using an ultrafiltration method with Centricon Plus-70 filter units of 30 KDa molecular-weight cutoff (Millipore Corporation, Billerica, MA, USA). The ultrafiltration process used a sterile glass pipette, where 70 mL of each wastewater sample was added into their correspondingly labeled sample filter cup pre-assembled with the filtrate collection cup. Each assembly was then sealed with a cap. The Centricon Plus-70 assemblies were placed into a swinging bucket rotor and centrifuged at 3000×g for 30 min at 20 °C. Subsequently, the filtrate was discarded, and the remaining 70 mL of the samples was added into their correspondingly labeled sample filter cup pre-assembled with the filtrate collection cup. Samples were spun at the same speed and temperature for 45 min. After centrifugation, the sample filter cup was separated from the filtrate collection cup. The concentration collection cup was then turned upside down and placed on top of the sample filter cup. The device was carefully inverted and placed into the centrifuge. Centricon Plus-70 filter units were centrifuged at 800×g for 2 min at 20 °C. After this step, the concentrated sample was collected from the concentration cup via a micropipette. The final volume was measured for each wastewater sample. If needed, 10 mM Tris–HCl, pH 8.5 buffer (Qiagen Sciences, Maryland, MD) was added to the concentrate to make up a total volume of 250 µL. If the final volume of the concentrate was over 250 µL, Tris buffer was not added. Aliquots containing 250 µL were made and stored at 4 °C and processed within 24 h.

Sludge cake preparation for ultrafiltration

To remove cells and virus particles from the SC samples, a 1X phosphate-buffered solution (PBS) with 0.15M NaCl, 0.05% Tween-20, and pH 7.5 was used. Approximately 30 g of SC sample per sampling event (Events 3 and 4) was collected and divided into six Falcon tubes for each event (∼5–6 g per tube). Approximately 30 mL of PBS was added to each tube. The Falcon tubes filled with SC samples were homogenized at constant agitation for 15 min at 2500 rpm in a vortex mixer. These tubes were then centrifuged at a speed of 4500×g for 50 min. The supernatant from each tube was subsequently recovered and transferred to a new sterile Falcon tube. For each sample event, 140 mL of supernatant was used for ultrafiltration as described previously.

Nucleic acid (DNA/RNA) extraction and fluorometric assessment

Once the final volume of concentrate was collected from each wastewater sample, the sample was pretreated with InhibitEX buffer (Qiagen Sciences, Maryland, MD) as indicated by the manufacturer. Then, QIAamp MinElute virus spin kit (Qiagen Sciences, Maryland, MD) was used to extract total nucleic acids from each wastewater sample as per the manufacturer’s instructions, which included the use of Qiagen Protease and carrier RNA (Qiagen Sciences, Maryland, MD). Samples were eluted in 75 µL of Buffer AVE (Qiagen Sciences, Maryland, MD), quantified, and stored at −80 °C for downstream processes. The nucleic acid concentration and purity were assessed using Qubit dsDNA high sensitivity and RNA assay kits in a Qubit 4 fluorometer (Invitrogen, Carlsbad, CA, USA). Qubit results can be found in Supplementary Materials (Table S1).

Quantitative PCR primers, probes, and gblocks gene fragments

Table 1 summarizes the primers and probes used in this study. Forward and reverse primers listed in Table 1 were used in the Primer-BLAST tool to extract gene target regions (Ye et al., 2012). Extracted regions were then uploaded to the Geneious software to verify oligonucleotide sequences associated to the flanking regions and probe. The generated sequences were sent to Integrated DNA Technologies (IDT, Inc., Coralville, Iowa, USA) to generate the desired gBlocks constructs. IDT manufactured all the primers used for quantitative PCR (qPCR) and quantitative reverse transcription PCR (RT-qPCR), as well as the probes Ast-P, Ring1a.2, and Ring 2.2 (Table 1). However, probes SaV124TP, SaV5TP, Tampere NSP3, AdV-P, PMMV-Probe, and 056P1 were manufactured by Life Technologies (Carlsbad, CA, USA).

Table 1 Primers and probes used in the present study.

Target	DNA or RNA	Primer/ Probe	Sequence (5′-3′)	Genomic target	References	
Adenovirus 40/41	DNA	AdV-F	GCC TGG GGA ACA AGT TCA G	Hexon	Molecular Microbiology & Genomics Team, British Columbia Centre for Disease Control (2017a)	
AdV-R	ACG GCC AGC GTA AAG CG	
AdV-P (Probe)	NED-ACC CAC GAT GTA ACC AC-MGB-NFQ	
Astrovirus	RNA	Ast-F	AAG CAG CTT CGT GAR TCT GG	Junction of polymerase and capsid	Molecular Microbiology & Genomics Team, British Columbia Centre for Disease Control (2017a)	
Ast-R	GCC ATC RCA CTT CTT TGG TCC	
Ast-P (Probe)	Cy5-CAC AGA AGA GCA ACT CCA TCG CAT TTG-Tao-IBDRQ	
crAssphage	DNA	056F1	CAG AAG TAC AAA CTC CTA AAA AAC GTA GAG	Genomic base pair region: 1,4731 bp–1,4856 bp	Stachler et al. (2018)	
056R1	GAT GAC CAA TAA ACA AGC CAT TAG C	
056P1	FAM-AAT AAC GAT TTA CGT GAT GTA AC	
Escherichia coli	DNA	784F	GTG TGA TAT CTA CCC GCT TCG C	uidA	Frahm & Obst (2003)	
866R	AGA ACG GTT TGT GGT TAA TCA GGA	
EC807 (Probe)	FAM-TCG GCA TCC GGT CAG TGG CAG T-BHQ1	
GI	RNA	COG1F-flap	AATAAATCATAACGYTGGATG CGNTTYCATGA	Norovirus GI	Kageyama et al. (2003); Wang et al. (2019)	
COG1R- flap	AATAAATCATAACTTAGACG CCATCATCATTYAC	
Ring1a.2 (Probe)	6-FAM- AGATYGCGR/ZEN/ TCYCCTGTCCA -IBFQ	Molecular Microbiology & Genomics Team, British Columbia Centre for Disease Control (2017b)	
GII	RNA	COG2F- flap	AATAAATCATAACARGARBCNA TGTTY AGRTGGAT GAG	Norovirus GII	Kageyama et al. (2003); Wang et al. (2019)	
COG2R-flap	AATAAATCATAATCGACGCCAT CTTCATTCACA	
Ring 2.2 (Probe)	JOE - TGGGAGGGY/ZEN/ GATCGCAATCT - IBFQ	Molecular Microbiology & Genomics Team, British Columbia Centre for Disease Control (2017b)	
Pepper Mild Mottle Virus	RNA	PMMV-FP	GAG TGG TTT GAC CTT AAC GTT TGA	1,878 bp–1,901 bpa and 1,945 bp– 1,926 bpa	Rosario et al. (2009)	
PMMV-RP	TTG TCG GTT GCA ATG CAA GT	
PMMV- Probe	FAM-CCT ACC GAA GCA AAT G-MGB-NFQ	
Rotavirus Type A	RNA	Tampere NSP3-F	ACC ATC TWC ACR TRA CCC TCT ATG AG	Non-structural Protein 3	Zeng et al. (2008)	
Tampere NSP3-R	GGT CAC ATA ACG CCC CTA TAG C	
Tampere NSP3 (Probe)	VIC-AGT TAA AAG CTA ACA CTG TCA AA-MGB-NFQ	
Sapovirus	RNA	SaV1F	TTG GCC CTC GCC ACC TAC	Junction of polymerase and capsid	Oka et al. (2006)	
SaV5F	TTT GAA CAA GCT GTG GCA TGC TAC	
SaV124F	GAY CAS GCT CTC GCY ACC TAC	
SaV1245R	CCC TCC ATY TCA AAC ACT A	
SaV124TP (Probe)	FAM-CCR CCT ATR AAC CA-MGB-NFQ	
SaV5TP (Probe)	FAM-TGC CAC CAA TGT ACC A-MGB-NFQ	
Notes.

a Corresponding nucleotide position of GenBank accession number M81413 (PMMV strain S).

Quantitative PCR assays

Taqman Environmental Master Mix 2.0 (Life Technologies, Carlsbad, CA, USA) was used for qPCR assays involving DNA enteric viruses and uidA. Taqman Fast Virus 1-Step Master Mix (4X) (Life Technologies, Carlsbad, CA, USA) was used for RNA enteric viruses via RT-qPCR. Each 10 µl qPCR and RT-qPCR reaction contained 500 nM of each of the forward primer and the reverse primer and 250 nM of its designated probe when targeting both DNA and RNA viruses. A total of 5 µl of Environmental Master Mix was utilized in each qPCR reaction for targeting DNA viruses, while 2.5 µl of 4×Fast Virus Master Mix was used in each RT-qPCR reaction for targeting RNA viruses. The uidA qPCR reaction consisted of  5 µl of Environmental Master Mix, 400 nM of each primer, and 100 nM of probe. All qPCR and RT-qPCR reactions used 2 µl of template.

Each qPCR and RT-qPCR reaction were performed in triplicate on an ABI QuantStudio 5 PCR system (Applied Biosystems, Foster City, CA, USA). The DNA enteric viruses (AdV and crAssphage) and uidA were subjected to the following conditions: 50.0 °C for 2 min and 95.0 °C for 10 min followed by 40 cycles of 95.0 °C for 15 s and 60.0 °C for 1 min. The RNA enteric viruses (SaV, RoV, AstV, GI and GII NoV, PMMV) were subjected to the following conditions: 50.0 °C for 5 min and 95.0 °C for 20 s followed by 40 cycles of 95.0 °C for 3 s and 60.0 °C for 30 s. Raw qPCR and RT-qPCR output files can be found on GitHub (https://git.io/J8VJ6).

Assessment of ultrafiltration for viral recovery efficiency

Armored RNA (Asuragen, Inc., Austin, TX, USA), an artificial virus packed with a 1000-bp single-stranded fragment and encapsulated in a protein coat, was used to assess recovery efficiency of the ultrafiltration method employed herein. We spiked in 16,000 copies of Armored RNA into 7.5 mL of representative RS, AS, and EF samples from the NESTP. For the SC sample, 1.25 g of solid SC was dissolved in 7.5 mL of PBS 1x then homogenized by vortexing at 2500 rpm for 15 min and centrifuged at 4500 x g for 50 min. The supernatant was transferred to a new Falcon tube to be undergoing the same treatment as the RS, AS, and EF samples. The 7.5-mL MilliQ negative control also spiked with 16,000 copies of Armored RNA. These five samples were first filtered through cheesecloth. 0.5 mL was aliquoted from each filtrate for subsequent assessment of recovery efficiency. The remaining volumes were subject to ultrafiltration using the Amicon Ultra-15 Centrifugal Filter Unit (Millipore Corporation, Billerica, MA, USA). Again, 0.5 mL of each flowthrough was stored for efficiency evaluation. Nucleic acid extraction of the retentate was performed in a manner similar to that described above. The final elution volume was 30 µL.

Primers (381F: 5′- AGCCTGTCAATACCTGCACC-3′and 475R: 5′- CACGCTTAGATCTCCGTGCT-3′),and probe (420P: 5′Cy5-AGAGTATGAGAGGTCGACGA-TAO 3′) were designed using Primer design tool of Geneious Prime version 2021.1.1 (http://www.geneious.com/) and targeted a 95-bp region within the Armored RNA genome. This targeted 95-bp fragment was sent to Integrated DNA Technologies (IDT, Inc., Coralville, Iowa) to synthetize a gBlock construct. Serial dilutions of this synthetic fragment were used to generate standards and quantify gene copy numbers (GCNs) of Armored RNA via RT-qPCR. DNA quantification using the same 95-bp fragment was also performed via qPCR. Standards, samples, and non-template controls were run in triplicate.

Thermal cycling reactions were performed at 50 °C for 5 min, followed by 45 cycles at 95 °C for 10 s and 60 °C for 30 s on a QuantStudio 5 Real-Time PCR System (Life Technologies, Carlsbad, CA, USA). For RNA assays, each 10-µl RT-qPCR mixture consisted of 2.5 µL 4X TaqMan Fast Virus 1-Step Master Mix (Life Technologies, Carlsbad, CA, USA), 400 nM each primer, 200 nM probe, and 2.5 µl of template, as well as ultrapure DNAse/RNAse free distilled water (Promega Corporation, Fitchburg, WI, USA). For DNA assays, 5.0 µL Master Mix was used.

Assessment of gene copy numbers by volume and biomass

Gene copy numbers (GCNs) were expressed in terms of sample (per mL or g of sample) and biomass (per ng of DNA or RNA). GCNs per mL of sample were calculated as previously described by Ritalahti et al. (2006). When calculating GCNs per mL of sample, the final volume recovered after filtering 140 mL of wastewater sample was used in the formula. For the SC samples, the mass of SC collected was used in the formula to produce results in GCNs per g of sample.

Collection of metadata for sampling events

To perform Principal Component Analysis (PCA) and Spearman’s rank correlation analysis for EF samples, metadata pertinent to the sampling events was retrieved. Water quality parameters obtained from the NESTP were combined with their October 2019 monitoring data (City of Winnipeg, Water and Waste Department, 2019) to complete some of the missing fields. For each value not found in either document, data interpolation was performed by taking an average of the corresponding values for the days before and after the sampling event. In addition, the Government of Canada’s historical weather database was utilized to obtain the mean temperature on the sampling dates and the total precipitation over three days before each sampling event (hereafter referred to as “precipitation”) (Environment and Climate Change Canada, 2021). The values for all parameters were transformed using log10, except for precipitation due to the presence of zero values. These variables were used with log10-transformed GCNs per mL sample for AdV, crAssphage, PMMV, and uidA (targets with quantifiable qPCR and RT-qPCR readings for all replicates across all events) as input for downstream analyses (PCA and Spearman’s rank correlation analysis).

Data handling, statistical analysis, and data visualization

Various applications were employed to process data at different steps of the pipeline. Input data, such as output from the quantitative PCR instrument, was subjected to manual formatting and cleaning in Microsoft Excel, which was also used to calculate GCNs per mL or g sample and per ng nucleic acid. GCNs and metadata were transformed using log10 function for analysis. GCNs across sampling events were pooled, and then comparisons were conducted across treatment.

R (R Core Team, 2021) and its integrated development environment RStudio (RStudio Team, 2021) as well as Statistical Analysis System (SAS, version 9.4 for Windows) were utilized to further process the data and perform statistical analyses and output visualizations. These operations included general linear models and multiple comparison procedures using Tukey’s tests, PCA (corresponding biplots were created using the package ggbiplot version 0.55 (Vu, 2011), and Spearman’s correlation matrix using the package Hmisc version 4.5-0 (Harrell Jr, 2021). The package reshape2 version 1.4.4 (Wickham, 2020) was used to reformat these correlation matrices to enhance compatibility with other data-handling tools. Information about other packages is provided in Supplementary Materials (Table S2). The R script used for analysis can be found on GitHub (https://git.io/J8VUl). Additional t-tests were conducted to compare water quality parameters between influent (RS) and discharges (EF).

Another software involved in data visualization was Tableau. Specifically, it was used to generate boxplots for GCNs per mL or g sample and per ng nucleic acid, as well as the heatmap representing the above-mentioned Spearman correlation matrix.

For all tests, a p-value of 0.05 was assumed to be the minimum level of significance.

Results

From our assessment of the sample processing method used in this study, the recovery efficiencies of Armored RNA as measured by RT-qPCR were between 14.03% and 15.94% for RS, 2.63−4.36% for AS, 12.36–18.74% for EF, and 2.40−5.45% for SC. Meanwhile, DNA recovery efficiencies were 32.48–40.87% (RS), 20.96–45.22% (AS), 14.14–20.15% (EF), and 23.41–68.42% (SC).

The GCN values for the DNA and RNA viruses and uidA were transformed into log10 form. These values were run through a general linear model Tukey-Kramer analysis, and the means of each wastewater processing stage for each target were analyzed. The GCNs were expressed in terms of volume (mL) or weight (g) of sample and biomass (ng of nucleic acids). The result for the GCN representing triplicate values from the corresponding sampling event was visualized as a circle in the box plots. We followed cut-off Ct values established by the Molecular Microbiology & Genomics Team at the British Columbia Centre for Disease Control (2017a; 2017b). With these values, the presence of DNA and RNA viral gene copies and uidA in the Milli-Q water (negative control) samples across all Events 1-4 were determined to be negative. The boxplots in Figs. 2–6 indicate the minimum, first quartile, median, third quartile, and maximum of the GCNs of each wastewater treatment sample across all events.

Overall, the average GCNs of the DNA and RNA enteric viruses assessed in RS, EF and SC were significantly and consistently higher (p < 0.05) compared to AS in terms of both volume and biomass (Figs. 2–3, Figs. S1 and S2). Average values of AdV ranged from 6.5 (AS) to 370.3 (SC) GCNs per ml or g of sample (Fig. 2A), while that RS and EF had on average 33.1 and 38.5 GCNs per ng DNA (Fig. 2B). Moreover, crAssphage ranged from 264 (AS) to 65,388 (SC) GCNs per ml or g of sample (Fig. 2C). Except for AS, crAssphage values in terms of biomass were unaltered across wastewater processes (Fig. 2D). In terms of RNA viruses, PMMV were observed in higher values (p-value ≤ 0.0001) in SC compared to other treatments (Fig. 3). Average values for SC were 6,478.1 copies per g and 189.2 copies per ng of sample. Similar to GCNs of AdV and crAssphage, lower values were observed in AS. On the other hand, values for uidA per volume were not significant (p-value ≥ 0.3716), we observed in higher and significant (p ≤ 0.0273) numbers of uidA per biomass in RS and EF compared to SC (Fig. 4). No other significant differences were detected for uidA. For all the aforementioned targets, there was a relatively higher number of gene copies observed in the EF across all events compared to AS samples.

Figure 2 Box plots of the number of gene copies of DNA enteric viruses across each wastewater stage throughout Events 1-4.

(A) and (C) visualize the number of gene copies per mL or g of sample, while (B) and (D) visualize the number of gene copies per ng of DNA. In (A, C, D) these quantities were log10-transformed for aesthetic purposes. The unit for the SC in (A) and (C) is gene copies per g of sample. Means with different letters indicate significant differences at the 0.05 level across treatments.

Figure 3 Box plots of the number of genes copies of PMMV across each wastewater stage throughout Events 1-4.

(A) visualizes the number of gene copies per mL or g of sample, while (B) visualizes the number of gene copies per ng of DNA. Both quantities were log10-transformed for aesthetic purposes. The unit for the SC in (A) is gene copies per g of sample. Means with different letters indicate significant differences at the 0.05 level across treatments.

Figure 4 Box plots of the number of gene copies of uidA across each wastewater stage throughout Events 1-4.

(A) visualizes the number of gene copies per mL or g of sample, while (B) visualizes the number of gene copies per ng of DNA. Both quantities were log10-transformed for aesthetic purposes. The unit for the SC in (A) is gene copies per g of sample. Means with different letters indicate significant differences at the 0.05 level across treatments.

NoV GI and GII were also targets for our study. Boxplots of their GCNs across the different wastewater stages and events 1-4 can be found in Supplementary Materials (Fig. S1). Norovirus GI was below RT-qPCR detection limits for all samples (RS, AS, and EF) during the events 1 and 2 (Fall season). Values were only detected during events 3 and 4 (Winter season). When detected NoV GI GCNs were higher (p-value <0.0001) in SC, EF and RS compared to AS. Values ranged between 0.4 (AS) to 51.8 (SC) per ml or g of sample, while that in terms of biomass NoV GI GCNs were observed between 0.05 (AS) and 2.1 (EF). In addition, NoV GII GCNs for all samples collected in Event 2 and AS samples in Events 3 and 4 (Winter season) were also below the detection limits (Fig. S1). Among the quantifiable samples, statistically significant GCN differences in terms of volume/mass and biomass were calculated for the pairs of AS-EF (p-values were 0.0129 and 0.0117, respectively), AS-RS (p-value =0.0223 per volume), and AS-SC (p-value <0.0001 for both). No other significant differences were detected among treatments for GCNs of NoV GII.

RoV gene copies across the various wastewater treatment stages from Event 1 to 4 were also examined. The boxplots illustrating these results in terms of both sample and biomass can be found in Fig. S2. RoV GCNs were only detected for wastewater samples collected during Events 3 and 4. Higher values per volume and biomass were detected in SC compared to the other treatments. RoV values ranged from 0.9 (AS) to 26.8 (SC) and 0.08 (AS) to 0.8 in terms of volume/mass and biomass, respectively. RoV GCNs were higher (0.0186) in EF per volume and biomass compared to AS. Looking at the EF-SC pair, the mean GCNs differed significantly in terms of volume/mass (p-value <0.0001) but not biomass (p-value = 0.8510). No other significant differences were detected for RoV per volume/mass or biomass.

In the present study, there was no detection of gene copies for AstV and SaV (Sav1, Sav124, and Sav5) in any of the wastewater samples across all events. In addition, to eliminate the possibility of inhibitors or contaminants such as humic acids, additional qPCR and RT-qPCR tests using bovine serum albumin (data not shown) were conducted with environmental samples (including AS). No significant differences were observed between samples with and without the enzyme.

To investigate any potential relationship between collected data for EF samples, PCA was performed with log10-transformed variables. We found that three components (PC1, PC2, and PC3) explained 99.14% of the variance between variables. A summary of the weight of components is included in the Supplementary Materials (Table S3). PC1 and PC2 were used to create the biplot in Fig. 5. Biplots for PC1 versus PC3 (Fig. S3) and PC2 versus PC3 (Fig. S4). Water quality parameters associated to UV-final effluents and used for PCA are summarized in Table 2. Physico-chemical and biological wastewater parameters (metadata) are also included in the Supplementary Materials. Water quality parameters available for RS and EF were also compared. Organic pollutants in EF from the NESTP were significantly reduced for BOD (p-value <,0.0001), COD (p-value <0.0001), sCOD (p-value = 0.0046), TOC (p-value <0.0001), TP (p-value = 0.0069), TS (p-value = 0.0223), and TSS (p-value <0.0001). No significant differences were observed between RS and EF for NH4-N (p-value 0.5006), and TN (p-value 0.3312).

Figure 5 Principal Component Analysis of log10-transformed EF parameters, PC1 versus PC2.

The only variable not log10-transformed was precipitation due to presence of zero values.

Figure 6 Heatmap showing Spearman’.s rank correlation analysis between parameters collected for EF sampling events.

Table 2 UV-treated final effluent water quality parameters measured in the North End Sewage Treatment Plant.c

	Sampling events			
Parameters	Oct-22-2019 (#1)	Nov-28-2019 (#2)	Dec-18-2019 (#3)	Feb-6-2020 (#4)‡	Mean	SD	
BOD (mg/L)	19.5a	13	18	26	19.13	4.64	
cBOD (mg/L)	4	5	7	7	5.8	1.5	
COD (mg/L)	51.5a	66	94	89	75.13	17.25	
sCOD (mg/L)	25	51	68	63	51.8	19.2	
E. coli (counts/100 ml)	60	60	90	1080	322.5	437.5	
Fecal coliforms (counts/100 ml)	100	20	40	640	200	295.3	
Grab Temperature (°C)	13.4	14.1	14.1	12.7	13.6	0.7	
Mean (Ambient) Temperature (°C)	2.7	−6.6	−17.0	−17.1	−9.5	9.5	
NH4-N (mg/L)	5.9	26.3	34.3	34.9	25.4	13.5	
NOx-N (mg/L)	7.5	5.3	3.3	2.1	4.5	2.4	
pH	7.12	6.81	6.79	6.99	6.93	0.14	
PO4-P (mg/L)	1.3	2.6	1.2	1.0	1.5	0.7	
Precipitation (mm)a	4.8	0	1	1.6	1.85	1.8	
TN (mg/L)	15.4b	40.7	49.8	50.5	39.1	14.22	
TOC (mg/L)	19.8b	21.2	29.6	34.4	26.25	6.02	
TP (mg/L)	1.67b	3.43	1.69	1.69	2.12	0.76	
TS (mg/L)	1,065	886	818	598	841.8	193	
TSS (mg/L)	19.5b	6	10	18	13.38	5.58	
Turbidity (NTU)	12.25b	4.2	6.3	9.2	7.99	3.03	
grab filtered UVT (%)	56.7	52.1	45.8	46.9	50.4	4.4	
Notes.

BOD biochemical oxygen demand

cBOD Carbonaceous Biochemical Oxygen Demand

COD chemical oxygen demand

sCOD soluble chemical oxygen demand

NH4-N ammonium-nitrogen

NOx-N nitrogen oxides –nitrogen

PO4-P orthophosphate

TN total nitrogen

TP total phosphorus

TOC total organic carbon

TS total solids

TSS total suspended solids

UVT Ultraviolet transmittance

a Cumulative amount of rainfall over three days.

b Parameters measured the day before and the day after were averaged and used to estimate parameters of sample date.

c Modified from Jankowski et al. (2022).

Overall, based on the biplot of PC1 and PC2, samples from the four events were distinct from one another, as point clusters of the four events can be seen occupying different quadrants. PC1, explaining 54.9% of the observed variance, received a notable and positive contribution from COD, cBOD, BOD, and TOC. Strongly negative contributors to PC1 were mean temperature, grab filtered UVT, NOx-N, and TS. These observations were supported by subsequent Spearman’s rank correlation analysis (Fig. 6), as COD, cBOD, BOD, and TOC demonstrated strongly positive correlations with one another (rho ranging between 0.8000 and 0.9487) (p-value <0.005) and strongly negative correlations with mean temperature, grab filtered UVT, NOx-N, and TS (rho ranging between −1.000 and −0.8000) (p-value <0.005). PC2 explained 31.9% of the variance between sampling events and showed a strong contribution from crAssphage, uidA, and grab temperature. This observation was also supported by the Spearman’s rank correlation analysis showing these variables having strongly positive correlation with one another (rho ranging between 0.7169 and 0.9218) (p-value <0.0100). Additionally, in the biplot, the axes representing E. coli and fecal coliform specifically pointed towards the same quadrant, which was reflected in their moderately positive Spearman’s coefficient (0.6325) (p-value = 0.0273). However, it is worth noting that uidA and E. coli exhibited a moderately weak negative correlation (rho = −0.3073), although it was not statistically significant (p-value = 0.3313). The two parameters with the strongest contribution against PC2 were grab pH and turbidity, which was illustrated by the strongly positive Spearman’s coefficient heatmap (rho = 0.8000) (p-value = 0.0018).

Discussion

The ultrafiltration method used in this study was assessed and the recovery efficiencies among all samples for Armored RNA were estimated to be between 2.40–18.74% for RNA. This range was comparable to other methods to concentrate viral particles such as JumboSep (13.38% ± 9.11%) or skimmed milk flocculation (15.27% ± 3.32%), spiked-in wastewater samples, and using Armored RNA as internal control (Yanaç et al., 2021). Viral particles may have been sorbed to biosolids present in wastewater samples, which were filtered out during the processing stage. In this context, matrix has a significant effect for recovery of viral particles. When compared to other environmental matrices such as surface water samples, recovery efficiency is higher using ultrafiltration (tangential flow filtration) (32.6% ± 11.81%) and skimmed milk flocculation (42.64% ± 15.12%) (J Francis and M Uyaguari, 2021, unpublished results). Water with high turbidity and amounts of suspended solids tend to saturate filters and impact the recovery of viral particles (Aslan et al., 2011; Karim et al., 2009; Uyaguari-Diaz et al., 2016). Additionally, the flow-through from ultrafiltration is another potential source of lost nucleic acid.

The GCNs were expressed in terms of biomass and volume (except for SC, which was expressed in g of sample). The higher abundance and more stable signal over time of GCNs of AdV and crAssphage (Fig. 2) as well as PMMV (Fig. 3) relative to the results of other assays make these target more representative for conducting comparisons with E. coli. This persistent presence is consistent with various longitudinal studies previously performed (Ballesté et al., 2019; Farkas et al., 2018; Farkas et al., 2019; Hamza et al., 2019; Nour et al., 2021; Schmitz et al., 2016; Tandukar, Sherchan & Haramoto, 2020; Worley-Morse et al., 2019; Wu et al., 2020).

A reduction of AdV, crAssphage, PMMV, and uidA GCNs was observed consistently in AS samples (Figs. 2–4). This could be a result of virus particles and bacterial cells being sorbed to larger fractions of organic matter that had been filtered by cheesecloth early in the sample-handling process or retained in the filtration devices as previously described. It is important to mention that samples were collected within a 2-hour period from RS → AS → EF consecutively within each sampling event. The higher GCNs of viruses and E. coli observed in the EF may be associated with the hydraulic retention time (12 h) in the facility and may not reflect wastewater treatment profiles at the time of collection (Rosman et al., 2014). In other words, the EF samples may not have been the corresponding RS samples collected earlier. The ideal situation would have seen the former being collected 12 h after the latter. It is best that similar logistical issues be accounted for in future studies. Other variables to consider are the overflow of sewage from rainy events and fluctuations in mixed liquor-suspended solids (Comber et al., 2019; Pérez et al., 2019). In our study, there were 4.6 mm of precipitation for Event 1, which may have affected the results. In the PCA analysis (Fig. 5), the vector for precipitation sharply denotes data points representing Event 1, indicating a possible relationship. Precipitation was also found to have positive correlations with grab flow (rho = 0.7746) and raw flow (rho = 0.7746) (Fig. 6). Nonetheless, further studies and/or more replications are needed to corroborate the potential link between precipitation and microbial counts.

Moreover, the duration of anaerobic sludge digestion is 25 days (City of Winnipeg, Water and Waste Department, 2020). In this context, GCNs of uidA in the SC were reduced by anaerobic digestion (Fig. 4). This may explain why the gene copies of uidA in terms of biomass were lower in SC compared to RS and EF (p-value <0.0273), but not when compared to AS (p-value = 0.0705). The average gene copies across all wastewater stages (RS, AS, and EF) for uidA were not significantly different in terms of both volume and biomass. When compared to uidA, enteric viruses were found to be at least one order of magnitude more abundant than the E. coli marker. Other studies have reported uidA in RS at copy numbers between 2 to 4 orders of magnitude higher that in our report (Jikumaru et al., 2020; Mbanga et al., 2020). In a related study conducted in the NESTP, we observed the same orders of magnitude for uidA gene marker across wastewater treatments (Jankowski et al., 2022) as the ones reported here. Although we acknowledge some sample may have been lost during the cheesecloth pre-filtration step, the same order of magnitude observed in both studies reflects true positive values for uidA across wastewater treatment processes.

GCNs of crAssphage in terms of biomass in SC were significantly higher than AS (p-value = 0.0123) (Fig. 2). For PMMV, SC samples had significantly more GCNs in terms of volume/mass and biomass than samples from other parts of the wastewater treatment process (p-value ≤ 0.0030) (Fig. 3). Since SC is the by-product of RS and AS using anaerobic digestion, this may indicate that the presence of crAssphage and PMMV was lower in the wastewater being treated in the AS, but higher in the solids. On the other hand, GCNs of AdV in terms of biomass were not significantly different between the AS and SC samples (Fig. 2B). Meanwhile, plant viruses such as PMMV remain more stable (in terms of biomass) during these digestion processes (Jumat et al., 2017).

The higher presence (p-value <0.0001) of RoV gene copies in the EF and SC in terms of volume/mass and biomass during the winter season (Fig. S2 and Table 2) may indicate a higher risk of transmission during cold seasons (Atabakhsh et al., 2020), since a greater presence of RoV in EF and SC has been previously found during the colder months of the year (Li et al., 2011) and little decay occurs at the desiccation step (Sánchez & Bosch, 2016).

The negative results of SaV (Sav1, Sav124, and Sav5) across all wastewater treatment stages during the fall and winter season are consistent with Varela et al. (2018) where samples were retrieved from a wastewater treatment plant in Tunisia. Their results did not support the general belief that the peak of detection of SaV occurs during the cold and rainy months of the year. However, quantitative detection of SaV in wastewater and river water in Japan showed an increased concentration of SaV in influents between winter and spring (December to May), but a decrease in SaV concentration during the summer and autumn months (July to October) (Haramoto et al., 2008). Yet another pattern of SaV presence was reported in France, as Sima et al. (2011) found the virus to be readily detected in influents but had no clear variations in numbers over the 9-month (October to June) duration of the study. Similarly, seasonal differences in SaV concentrations were not statistically significant in a 3-year study conducted by Song et al. (2021) in China between 2017 and 2019. As a result, there are other factors that can influence wastewater SaV concentrations. For example, it has been hypothesized that isoelectric point could affect how viruses and their different strains behave in bioreactors (Miura et al., 2015). Monitoring over a time period longer than our current study would likely shed more insight into the seasonal variation in the presence of SaV in wastewater.

The gene copies of NoV GI and GII were below the detection limit in many of the AS samples (in terms of both volume and biomass), but still relatively abundant in the other samples (Fig. S1). A possible explanation for the greatly reduced viral GCNs in AS samples is the high efficiency with which NoV GI and GII are removed, a notion supported by literature (Ibrahim et al., 2020; Kitajima et al., 2014; Schmitz et al., 2016). Furthermore, considering the observation that these viruses were found in abundance in SC samples, another contributing factor could be limitations in the sample collection process, which might not have adequately retrieved the slurry part of the sludge where the viruses are found in greater numbers as they might have sorbed to the larger fractions of the sludge solids. The relative abundance of NoV GI and GII gene copies in RS, EF, and SC during the colder months (December and February) and the absence of NoV GII in RS in November may be due to seasonal variability including intermittences (Guix et al., 2002; Pérez et al., 2019; Sánchez & Bosch, 2016). In this context, the presence of NoV GI and GII gene copies in RS during Events 3 and 4 (December and February) is consistent with a study conducted by Flannery et al., (2012), in which the concentration of NoV GI and GII gene copies in the influents of a wastewater treatment plant were significantly higher during the winter months (January to March). During colder months, we observed mean (ambient) temperatures averaged −17.05 °C (Table 2). This seasonal trend is also reflected colloquially through the virus’s sobriquet, the winter vomiting bug (Farkas et al., 2021). Overall, GCNs of NoV GI and GII did not seem to be reduced by the wastewater treatment process (Fig. S1).

High numbers of AstV gene copies (per liter) in sewage samples from the Greater Cairo area in Egypt were observed at the end of autumn (daily mean temperatures ranged from 10.8 °C to 22.8 °C) and during the winter months (daily mean temperatures ranged from 6.8 °C to 17.0 °C) (El-Senousy et al., 2007; Weather Spark, 2022), but the AstV concentrations tended to decrease as temperatures increased. These results are different from our findings where there was no detection of AstV in any of the wastewater treatment stages across all events. Mean (ambient) and grab sample average temperatures during our study-period were −9.5 °C (range 2.7 °C to −17.1 °C) and 13.6 °C (range 12.7 °C to 14.1 °C), respectively (Supplementary Materials and Table 2). These results may be due to seasonal variability (Pérez et al., 2019) as well as reflect the pattern of infection within the community under study (Corpuz et al., 2020).

Grab filtered UVT being inversely correlated with COD, cBOD, BOD, and TOC is consistent with the widespread use of UV radiation to regulate microbial growth in a variety of medium, including water (Raeiszadeh & Adeli, 2020). Furthermore, it had been suggested that UV is an important influence to the survival of pathogens in wastewater environments, especially in cold weather conditions, such as those found in Manitoba during the surveying period (Murphy, 2017). The NESTP uses UV as disinfection treatment and significantly reduces most organic pollutants (RS vs EF). In the present study, we observed an incomplete removal of nitrogen (TN and NH4-N). Excess of these nutrients is associated to efficiency of the activated sludge process and discharges can be toxic for aquatic organisms and/or cause algal blooms (Templeton & Butler, 2011; Chahal et al., 2016). Moreover, virus-mediated transfer of nitrogen can occur from heterotrophic bacteria such as the pool of microorganisms present in AS to primary producers in aquatic environments (Shelford & Suttle, 2018; Chen et al., 2021). Further studies are needed to evaluate the survival of enteric viruses in these reservoirs by using modification to the biological treatment, the disinfection process and/or physical methods (i.e., filtration methods). Some of these modifications may include fixed bed reactors (Sizirici & Yildiz, 2020), biofilm systems such as membrane bioreactors, biofilters, biofiltration, and carriers (Zhao et al., 2019). Other disinfection processes include the use of chlorine (liquid sodium hypochlorite solution, solid calcium hypochlorite) or newer methods such as ozone (Mezzanotte et al., 2007; Abou-Elela et al., 2012; Collivignarelli et al., 2018). Although microfiltration and ultrafiltration can be used to reduce bacterial and protist pathogens, and enteric viruses (Chahal et al., 2016), membrane foulants and fouling mechanisms occur in WWTP effluents (Nguyen, Roddick & Harris, 2010).

There is a possibility that viral GCNs quantified in the EF may represent an overestimation of the actual number of infectious viral particles since quantitative PCR detects both infective and non-infective agents and UV treatment influences viral viability (Lizasoain et al., 2017). Thus, the interpretation of these results must be performed with caution. Future studies could incorporate culturable assays for a more complete and accurate evaluation as well as longer time series. On the other hand, it is also possible that the non-enveloped enteric viruses (Corpuz et al., 2020) studied here survived the wastewater treatment process. Non-enveloped viruses are more resilient than their enveloped counterparts in numerous environmental conditions and water treatment processes (La Rosa et al., 2020). This is because of the latter group’s envelope, which contains receptors needed for infection; if the envelope is lysed, infection is not possible (La Rosa et al., 2020). Various publications have noted the resilience of non-enveloped viruses after wastewater treatment (Adefisoye et al., 2016; Campos & Lees, 2014; Farkas et al., 2019; Fitzgerald, 2015; Fong, Xagoraraki & Rose, 2010; Li et al., 2021; Prevost et al., 2015; Ruggeri et al., 2015; Varela et al., 2018). In this context, we have consistently detected GCNs of AdV, crAssphage, and PMMV in environmental surface waters receiving discharges from the NESTP, two other WWTPs, and other areas radiating away from the WWTPs within the City of Winnipeg (J Francis and M Uyaguari, 2021, unpublished data). Therefore, despite potential factors affecting interpretation, our results still reflect the presence of several non-enveloped enteric viruses in EF and SC samples with reasonable quantitative accuracy. While it is undeniable the key role that WWTPs have played in reducing nutrient loading and maintain water quality, they are not designed to remove all microbes, especially smaller fractions such as viruses, genetic material or micropollutants. In this aspect, unproperly treated effluents or mismanagement of SC, often employed as fertilizers, may represent a route for microbes including pathogens and their genetic material to be transported into downstream waterways or introduced into other settings (i.e., urban → agricultural, urban → rural).

Conclusion

Our study’s primary goal was to identify human enteric viruses with the potential to become alternative indicators of fecal pollution. Towards that end, we found AdV, crAssphage, and PMMV as more stable viral indicators of water quality due to their quantifiability illustrated in this investigation and the literature. RNA viruses such as NoV GI and GII, and Rotavirus were detected during colder months, while AstV and SaV could not be detected in any of the samples. Regular monitoring of these organisms can be useful complements to current methods for assessing wastewater treatment processes including seasonal viruses. Wastewater surveillance of SARS-CoV-2 during the current pandemic has demonstrated the utility of early warning tools. Such vigilance could be a helpful tool to assist public health efforts in the event of a viral outbreak.

Our study indicated that enteric viruses may have survived the wastewater treatment process and viral-like particles are possibly being released into the aquatic environment. Therefore, in addition to such methods as UV radiation (which is currently used in the NESTP and was shown in our study to be inversely correlated with biological parameters), we also suggest that WWTPs consider implementing modifications and/or additions (disinfection processes) to their workflow to reduce the number of viral particles being released into the aquatic environment.

Supplemental Information

Supplemental Information 1 Qubit results of extracted nucleic acid samples

Click here for additional data file.

Supplemental Information 2 R packages used that were not mentioned in the manuscript

Click here for additional data file.

Supplemental Information 3 Summary of weight of components of PCA for EF samples and related metadata

Click here for additional data file.

Supplemental Information 4 Box plots of the number of gene copies of Noroviruses GI and GII across each wastewater stage throughout Events 1-4

The unit for the SC in Figs. S1A, S1C is gene copies per g of sample. In S1C this quantity was log10-transformed for aesthetic purposes. Means with different letters indicate significant differences at the 0.05 level across treatments.

Click here for additional data file.

Supplemental Information 5 Box plots of the number of gene copies of Rotavirus across each wastewater stage throughout Events 1-4

The unit for the SC in Fig. S2A) is gene copies per g of sample. Means with different letters indicate significant differences at the 0.05 level across treatments.

Click here for additional data file.

Supplemental Information 6 Principal Component Analysis of log10-transformed EF parameters, PC1 versus PC3

The only variable not log10-transformed was precipitation due to presence of zero values.

Click here for additional data file.

Supplemental Information 7 Principal Component Analysis of log10-transformed EF parameters, PC2 versus PC3

The only variable not log10-transformed was precipitation due to presence of zero values.

Click here for additional data file.

Special thanks to the City of Winnipeg and Palwinder Singh, graduate student, Department of Civil Engineering at the University of Manitoba for sample collection. This research was conducted at the University of Manitoba. We would like to acknowledge that the University of Manitoba campuses are located on original lands of Anishinaabeg, Cree, Oji-Cree, Dakota, and Dene peoples, and on the homeland of the Métis Nation.

Abbreviations

AdV Adenovirus

AS activated sludge

AstV Astrovirus

BOD biochemical oxygen demand

cBOD carbonaceous biochemical oxygen demand

COD chemical oxygen demand

EF effluents

GCN gene copy number

NESTP North End Sewage Treatment Plant

NH4-N ammonium-nitrogen

NoV Norovirus

NOx-N nitrogen oxides - nitrogen

PCA Principal Component Analysis

PMMV Pepper mild mottle virus

PO4-P orthophosphate as phosphorus

RoV Rotavirus

RS raw sewage

qPCR quantitative PCR

RT-qPCR quantitative reverse transcription PCR

SaV Sapovirus

SC sludge cake

sCOD soluble chemical oxygen demand

TN total nitrogen

TOC total organic carbon

TP total phosphorus

TS total solids

TSS total suspended solids

uidA β-d-glucuronidase gene

WWTP wastewater treatment plant

Additional Information and Declarations

Competing Interests

Author Contributions

Data Availability

The authors declare there are no competing interests.

Audrey Garcia and Tri Le performed the experiments, analyzed the data, prepared figures and/or tables, authored or reviewed drafts of the paper, and approved the final draft.

Paul Jankowski performed the experiments, prepared figures and/or tables, authored or reviewed drafts of the paper, and approved the final draft.

Kadir Yanaç performed the experiments, authored or reviewed drafts of the paper, and approved the final draft.

Qiuyan Yuan conceived and designed the experiments, authored or reviewed drafts of the paper, and approved the final draft.

Miguel I. Uyaguari-Diaz conceived and designed the experiments, analyzed the data, prepared figures and/or tables, authored or reviewed drafts of the paper, and approved the final draft.

The following information was supplied regarding data availability:

The Raw data and scripts are available at Github: https://git.io/JRmfe.

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
