# Peer review of "Quantification of human enteric viruses as alternative indicators of fecal pollution to evaluate wastewater treatment processes"

_PeerJ, doi:10.7717/peerj.12957_

## Round 0.1 · original submission · Major Revisions

Your study is well written and of interest to Environmental science especially to water quality monitoring. However, as pointed out by the reviewers, more replicates have to be done, viral capture efficiencies need to be evaluated for all samples and statistical analyses revised. I agree with all the concerns raised by the reviewers and in order to accept your manuscript all of them should be addressed. I will be happy to receive a new version with a point-by-point response letter to reviewers. Thank you.

Reviewer 1 ·

Basic reporting

There were no major issues with regard to basic reporting. I found the article clear for the most part with some grammatical errors in a few places:

Line 163-165 and Table 1: I believe you need to use the exact name of the primers and probes from the papers you got them. I don’t think you should change the name from what it is in the paper you cited. Some of the papers you cited have multiple primers and probes (and each primer and probe may have different efficiencies) for each organism and it would cause confusion on which primer or probe you are using.

Figures 2-4: What is the difference between a and b figures? Please explain more clearly in caption or titles.

Line 370: typo

Line 418: 'caution' instead of 'cautiousness'

Experimental design

I believe the article had several challenges when it comes to experimental design which would need to be fixed for publication. One of the main challenges is the lack of replication in the experimental design which makes the study less rigorous. Previous studies have looked at this topic using 5-10 samples whereas this study only used 2 samples per season. Replication is important in this field as concentrations are highly variable in wastewater samples. Hence, comparisons cannot be be accurately made without more replication. I also don't believe you can compare concentrations received from 2 different samples if you used different viral capture and extraction approaches. Especially if you have not determined the relative viral capture efficiencies associated with each approach. Further line by line comments are stated below:

Line 96: Was it an actual virus with genome and capsid that were spiked in or was it just the RNA genome? If it is only the RNA genome, this will only give you efficiency of collecting genomes not collecting viruses which have different sizes. Also it is an RNA target which might have different capture efficiency from a DNA target.

Line 97: Please clarify what exactly was not used in the study.

Line 114-116: Isn’t it possible that there are microbes on the solid waste or debris that was filtered by the funnel? Did you quantify the viruses on those? If so, please report those concentrations.

Line 135-143: What is the typical recovery from this process? Maybe you could cite a paper that discusses this.

Line 228: why was this only done for raw sewage and not activated sludge and effluent? Wouldn’t there be different efficiencies in the different types of samples? the efficiency associated with capturing viruses from SC is also very important.

Figures 2-6: how are you making box plots with duplicate samples from each event and treatment? Did you take more than two samples? Or are you using the qPCR data as replicates? Could you clarify what exactly are the data points used to make these box plots. Could you also state the limit of detection.

Line 330-333: Could also be resolved with more replication.

Line 343-344: Because the concentration methods are different, I don’t believe this (comparing between SC and RS) is a good comparison. Differences could be due to differences in extraction and concentration recovery efficiencies. I suggest doing recovery efficiencies for SC and all other treatment samples.

Line 352-355: You would need more sampling replicates than you have to compare between seasons.

Line 394-403: I think you have too few points for these analyses. Maybe these can be cut.

Line 376: Were there any culturable assays measured in this study? I think another limitation that can be stated is the use of only molecular assays which do not measure viability and can cause overestimation of risk.

Validity of the findings

Since this study does not have good replication, it greatly impacts validity of findings and comparisons in the discussion. More comments below:

Line 227-229 and 303-307: This recovery efficiency is pretty low. Is it because you pre filtered your sample?

Line 325-329: From your results, concentrations of indicators decreased in AS but then increased in EF after. How did this happen? Especially for viruses that cannot grow in wastewater. If your reasoning is the hydraulic retention time, could you explain that more.

Line 330-333: Could also be resolved with more replication.

Line 356-359: Possibly because in that country, the winters aren’t as cold as Winnipeg hence less viral transmission.

Figure 4: The concentration of E. Coli being lower than crass and Pmmv is very strange. Is this the usually concentration for this particular E. coli marker in raw sewage? If so, I believe other markers would be better such as EC 23S (https://doi.org/10.1021/es302222b) or even the culturable marker.

Additional comments

I believe if more replication is done and viral capture efficiencies are evaluated for all samples, the paper can be accepted.

Reviewer 2 ·

Basic reporting

Thank you for allowing me to review your manuscript “Quantitation of human enteric viruses as alternative indicators of fecal pollution to evaluate wastewater treatment processes”. This novel and well written article provides findings on important research into using human enteric viruses as indicators of contamination in wastewater.
I thought the manuscript used clear and professional language throughout. The introduction provided a clear outline of background information on usage of E.coli as a means of detection in wastewater but that only using E.coli excludes the possibility of the presence of other contaminants. Relevant prior literature and citations were included.
All figures were relevant to the content of the article and the raw data is easily accessible and all links such as those to the Github pages were accessible. The results were directly related to the main hypothesis and self-contained in the article. The main conclusion that AdV, crAssphage and PMMV be used as viral indicators of water quality was supported in the results section.

I do I have some minor comments on language errors I found throughout the manuscript as well as a question about a figure:
- Line 39-40: “This indicate that..” should be “indicated”
- Line 370 – “The NESTP” Should be removed
- The authors use the word “quantitated” throughout the manuscript. I would suggest using “quantified” instead.
- Figure S2B – It seems to be off the page or at the very edge of the page. Ensure that this is reasonable for publication.

Experimental design

The design of the experiment seems robust with proper techniques utilized and proper citations for certain methods.
I did have a couple questions related to statistics –
1. Using the Spearman rank correlation analysis was the average of all gene copies taken (average of event 1-4) and compared in this way? Do you think using this average value skewed results if there are differences between samples taken in the fall compared to winter?
2. Were any statistical tests carried out to compare samples collected in Fall vs Winter?
3. Line 207 – The authors discuss the statistical analysis. Please identify what you are actually comparing here and throughout the results. Are you comparing the average GCN of one sample type to another across each viruses? It was difficult to follow and determine what was statistically significant.

Validity of the findings

All the data is easily accessible, and findings are reliable. I thought the conclusion was well written and clearly outlined the end results. It is apparent that this work will benefit future work monitoring water quality.

---

## Round 0.2 · Major Revisions

Your work is important to the field and I will be happy to see it published. However the reviewer’s concerns have to be addressed. I believe they are reasonable and aimed to improve the manuscript. Since I understand replications may not be possible to obtain now, to overcome this issue it would be wise following the advice of pooling data. Please send a revised version of the manuscript and a point-by-point answer letter as soon as possible. Thank you very much

Reviewer 1 ·

Basic reporting

I just have a main challenge with 3 figures.

Figures 2-4: Are these replicates you used for making these box plots? please explain how you are making these box plots

Experimental design

Experimental design is still a challenge as you do not have replication and are making claims of significance. Specific comments below:

You did not report water quality parameters (pH, temperature, conductivity, etc) for all your samples, especially the liquid samples. These are needed to generalize results and for your comparisons between seasons. please report this data.

Previous studies have examined done this exact experiment and should be cited (https://doi.org/10.1016/j.envint.2019.105452 , https://doi.org/10.1016/j.watres.2019.02.042)

Line 413-416: what temperatures? can you report temps?

Line 409-412: I understand the challenge with getting replication but you cannot make some of these claims without replication especially when comparing significance between fall and winter seasons because you only had 2 samples for each season. you should just lump all four events together then make comparisons as you don't have enough samples to compare between events. Do not just change this for this section (NoV) but change for all other sections that you choose to make this comparison for

Validity of the findings

Your study is very crucial to developing each of these fecal indicators for water quality assessment. Please show this in your writing and state why this study is so novel compared to previous studies done in the past.

---

## Round 0.3 · Minor Revisions

I am ready to accept your manuscript, but there is an issue that needs to be addressed. Please indicate in the text when you mention the quantification of RNA viruses that it is by RT-qPCR, to differentiate it from qPCR to quantify DNA viruses. As it now stands it appears that both types of viral nucleic acids are quantified in the same way. Submit the edited manuscript as soon as possible and it will be accepted as long as these changes are made. Thanks

---

## Round 0.4 · accepted · Accept

Thanks for the quick response and for editing the manuscript. My concerns have been successfully addressed and it is now ready for acceptance. Congratulations on your work, I am sure it will advance the research and development of fecal indicators for montoring water quality assessment.